# Management and Survival of Elderly and Very Elderly Patients with Ovarian Cancer: An Age-Stratified Study of 1123 Women from the FRANCOGYN Group

**DOI:** 10.3390/jcm9051451

**Published:** 2020-05-13

**Authors:** Yolaine Joueidi, Ludivine Dion, Sofiane Bendifallah, Camille Mimoun, Alexandre Bricou, Krystel Nyangoh Timoh, Pierre Collinet, Cyril Touboul, Lobna Ouldamer, Henri Azaïs, Yohann Dabi, Cherif Akladios, Geoffroy Canlorbe, Pierre-Adrien Bolze, Hélène Costaz, Mathieu Mezzadri, Tristan Gauthier, Frederic Kridelka, Pauline Chauvet, Nicolas Bourdel, Martin Koskas, Xavier Carcopino, Emilie Raimond, Olivier Graesslin, Lise Lecointre, Marcos Ballester, Cyrille Huchon, Jean Levêque, Vincent Lavoué

**Affiliations:** 1Department of Gynecology, Rennes University Hospital, Hôpital Sud, 35000 Rennes, France; yolaine.joueidi@orange.fr (Y.J.); ludivine.dion@chu-rennes.fr (L.D.); krystel.nyangoh.timoh@chu-rennes.fr (K.N.T.); jean.leveque@chu-rennes.fr (J.L.); 2IRSET, Equipe 8, INSERM U 1085, 35000 Rennes, France; 3Service de Gynécologie, Hopital TENON, AP-HP, 75020 Paris, France; sofiane.bendifallah@aphp.fr (S.B.); cyril.touboul@aphp.fr (C.T.); 4Department of Gynecology and Obstetrics, Lariboisiere Hospital, 75010 Paris, France; camille.mimoun@aphp.fr; 5Croix Saint Simon, Service de Chirurgie Gynécologique, 75000 Paris, France; alex.bricou@gmail.com (A.B.); mballester@hopital-dcss.org (M.B.); 6Service de Gynécologie, Hôpital Jeanne de Flandres, 59000 Lille, France; Pierre.COLLINET@CHRU-LILLE.FR; 7Service de Gynécologie, CHU Tours, 37000 Tours, France; louldamer@yahoo.fr; 8Assistance Publique des Hôpitaux de Paris (AP-HP), Department of Gynecological and Breast Surgery and Oncology, Pitié-Salpêtrière University Hospital, 75013 Paris, France; henri.azais@aphp.fr (H.A.); geoffroy.canlorbe@aphp.fr (G.C.); 9Service de Gynécologie Obstétrique, Centre Hospitalier Inter Communal de Créteil, 94000 Créteil, France; yohann.dabi@gmail.com; 10Service de Gynécologie Obstétrique, CHU Hautepierre, 67000 Strasbourg, France; cherif.akladios@gmail.com (C.A.); lise.lecointre@chru-strasbourg.fr (L.L.); 11Service de Gynécologie Obstétrique, CHU Lyon Sud, Hospices Civiles de Lyon, 69000 Lyon, France; pierre-adrien.bolze@chu-lyon.fr; 12Département d’Oncologie Chirurgicale, Centre Georges François Leclerc, Unicancer, 21000 Dijon, France; hcostaz@cgfl.fr; 13Service de Gynécologie Obstétrique, Hôpital Lariboisière, APHP, 75000 Paris, France; matthieu.mezzadri@aphp.fr; 14Service de Gynécologie Obstétrique, CHU de Limoges, 87000 Limoges, France; Tristan.Gauthier@chu-limoges.fr; 15Service de Chirurgie Oncologique et Gynécologique, 4000 Liège, Belgique; frederic.kridelka@chu.ulg.ac.be; 16Service de Gynécologie Obstétrique, CHU de Clermont Ferrand, 63000 Clermont Ferrand, France; po.chauvet@gmail.com (P.C.); nicolas.bourdel@gmail.com (N.B.); 17Service de Gynécologie, Hôpital Bichat, APHP, 75018 Paris, France; martin.koskas@aphp.fr; 18Service de Gynécologie, Hôpital La Timone, APHM, 13000 Marseille, France; xcarco@free.fr; 19Service de Gynécologie, Hôpital Universitaire de Reims, 51000 Reims, France; eraimond@chu-reims.fr (E.R.); olivier.graesslin@gmail.com (O.G.); 20Department of Gynecology, Centre Hospitalier de Poissy, 78000 Poissy, France; cyrillehuchon@yahoo.fr

**Keywords:** ovarian cancer, elderly, surgery, chemotherapy, frailty

## Abstract

Elderly women with ovarian cancer are often undertreated due to a perception of frailty. We aimed to evaluate the management of young, elderly and very elderly patients and its impact on survival in a retrospective multicenter study of women with ovarian cancer between 2007 to 2015. We included 979 women: 615 women (62.8%) <65 years, 225 (22.6%) 65–74 years, and 139 (14.2%) ≥75 years. Women in the 65–74 years age group were more likely to have serous ovarian cancer (*p* = 0.048). Patients >65 years had more >IIa FIGO stage: 76% for <65 years, 84% for 65–74 years and 80% for ≥75 years (*p* = 0.033). Women ≥75 years had less standard procedures (40% (34/84) vs. 59% (104/177) for 65–74 years and 72% (384/530) for <65 years (*p* < 0.001). Only 9% (13/139) of women ≥75 years had an Aletti score >8 compared with 16% and 22% for the other groups (*p* < 0.001). More residual disease was found in the two older groups (30%, respectively) than the younger group (20%) (*p* < 0.05). Women ≥75 years had fewer neoadjuvant/adjuvant cycles than the young and elderly women: 23% ≥75 years received <6 cycles vs. 10% (*p* = 0.003). Univariate analysis for 3-year Overall Survival showed that age >65 years, FIGO III (HR = 3.702, 95%CI: 2.30–5.95) and IV (HR = 6.318, 95%CI: 3.70–10.77) (*p* < 0.001), residual disease (HR = 3.226, 95%CI: 2.51–4.15; *p* < 0.001) and lymph node metastasis (HR = 2.81, 95%CI: 1.91–4.12; *p* < 0.001) were associated with lower OS. Women >65 years are more likely to have incomplete surgery and more residual disease despite more advanced ovarian cancer. These elements are prognostic factors for women’s survival regardless of age. Specific trials in the elderly would produce evidence-based medicine and guidelines for ovarian cancer management in this population.

## 1. Introduction

Ovarian cancer is the seventh most common cancer in women (7.1/100,000 women) and the fourth cause of mortality by cancer in women. It accounts for an estimated 239,000 new cases and 152,000 deaths worldwide annually [1]. In France, its incidence is around 4700 new cases per year, and it is responsible for 3100 deaths [2]. The mean age at diagnosis is 65 years old with approximately 48% of patients older than 65 years [3], and outcomes generally worsen as the age of the patient increases [4]. Most patients with ovarian cancer are diagnosed at an advanced stage with a poor prognosis. Currently, although the 5-year relative survival rate for women with ovarian cancer increased from 36% in 1975–1977 [5] to 46% in 2005–2011 [6], it remains the most-deadly gynecological cancer.

Today, the standard treatment for ovarian cancer involves surgery and chemotherapy. To achieve complete cytoreduction, complex surgical procedures are necessary such as salpingo-oophorectomy with hysterectomy, omentectomy, pelvic lymphadenectomy and para-aortic lymphadenectomy. Bowel resection, pancreas resection, splenectomy, diaphragmatic stripping and partial liver resection may also be required and can lead to serious complications [7]. The benefit of this kind of treatment remains controversial in elderly patients considered frailer [8] and more prone to a higher rate of complications [9,10,11]. They are therefore often undertreated [12,13] in spite of the fact that some authors consider age to be an independent prognostic factor [14]. Data concerning the management of elderly patients with ovarian cancer are lacking since they have historically been under-represented in clinical trials [13,15,16]. As life expectancy increases (in the next few decades, approximately 20% of the world population will be over 65 years old [17]), the incidence of ovarian cancer in the elderly will rise. Oncologic management in this population is thus an emerging critical issue.

This study aimed to evaluate the management of young, elderly and very elderly patients with ovarian cancer in a large French multicenter cohort. We also studied overall survival (OS), disease-free survival (DFS), and cancer specific survival (CSS) rates.

## 2. Material and Methods

### 2.1. Patients

We collected data on women who had received surgical treatment for a histologically proven ovarian cancer between January 2000 and December 2016 from eight teaching hospitals in France (Creteil University Hospital, Jean Verdier University Hospital, Lille University Hospital, Poissy University Hospital, Tenon University Hospital, Tours University Hospital, Rennes University Hospital and Strasbourg University Hospital). All the data had been entered in a single ovarian cancer database. We excluded patients with no surgical treatment (thus, only chemotherapy patients were excluded), with non-epithelial ovarian carcinoma, benign tumors, and patients with unknown FIGO stage (International Federation of Gynecology and Obstetrics system (FIGO)). The research protocol was approved by the Institutional Review Board of the Collège National des Gynécologues et Obstétriciens Français (CEROG 2016-GYN-1003).

### 2.2. Data Collection

The following demographic and clinical data were collected: age, body mass index (BMI, calculated as weight in kilograms divided by the square of height in meters), parity and Ca125 at diagnosis. Preoperative assessment of surgical risk was performed according to the American Society of Anesthesiology (ASA) Physical Status Classification by the attending anesthesiologist at the time of surgery [18]. The type of surgical procedure, pathologic data (International Federation of Gynecology and Obstetrics system (FIGO) stage), nodal status and adjuvant therapy were also collected. Patients were divided into three cohorts: young women (aged < 65 years), elderly women (aged 65–74 years), and very elderly women (aged ≥ 75 years).

### 2.3. Histology

Tumors were classified according to their pathologic status. Histologic subtypes were classified as serous adenocarcinoma, endometroid, mucinous, clear-cell and mixed epithelial tumors. Patients were then graded according to the FIGO 2014 classification [19].

### 2.4. Treatment

Cytoreductive surgery performed before chemotherapy was called “primary debulking surgery,” and when performed after chemotherapy it was called “interval debulking surgery” or “surgery after 6 cycles.” The treatment sequence was determined on an individual basis according to the patient’s general condition, exam results (CT scan, MRI…), and after multidisciplinary concertation. Standard surgery for ovarian cancer consists of hysterectomy, salpingo-oophorectomy, omentectomy, pelvic and para-aortic lymphadenectomy. Bowel resection, pancreas resection, splenectomy, diaphragmatic stripping and partial liver resection may also be required to achieve complete cytoreduction.

The surgical procedures were classified according to Luyckx et al.’s publication [20]: Group 1 included standard surgery with hysterectomy, bilateral salpingo-oophorectomy, rectosigmoid resection, infragastric omentectomy, pelvic and aortic lymphadenectomy, and, when applicable, appendectomy; Group 2A included standard surgery plus relatively routine upper abdominal surgery such as stripping of the diaphragmatic peritoneum and splenectomy alone; and Group 2B included ultra-radical surgeries involving a combination of digestive tract resections (right colon and caecum, total colectomy, and others), organ resection (spleen, gallbladder, partial gastrectomy, and others), celiac lymph node dissection, and total abdominal peritoneum stripping in addition to standard surgery.

We evaluated the complexity of the surgical procedure using Aletti’s surgical complexity score (SCS) [3]. This score classifies the surgical complexity including associated procedures in three groups: low (≤3), intermediate (4–7) and high (≥8).

Surgery complications were assessed according to the Clavien-Dindo classification [21] as minor (grades I and II) or major (grades III, IV and V). Minor complications included anemia, occlusion, wall abscess, pneumonia, lymphocele, thrombosis, and major complications were lymphocele puncture, pleural effusion drainage, pulmonary embolism, organ failure, sepsis, hemorrhage, requirement for repeat surgery, or death.

Adjuvant therapies included chemotherapy and bevacizumab. Chemotherapy was based on platinum salts (CARBOPLATINE AU5 or AU6) and taxanes (PACLITAXEL 175 mg/m^2^) every 3 weeks with at least 6 cycles. The chemotherapy protocol or bevacizumab administration was modified on an individual basis after multidisciplinary concertation.

### 2.5. Follow-Up Assessment

Clinical follow-up consisted of physical examinations and the use of imaging techniques according to the clinical findings during visits conducted every 3 months for the first 2 years, every 6 months for the following 3 years, and once a year thereafter.

### 2.6. Outcome Measures

The outcome measures were OS (overall survival), DFS (disease free survival), and CSS (cancer specific survival) rates calculated from date of recurrence, date of death, and date of cancer-related death. We also analyzed the rates and types of Clavien-Dindo complications.

### 2.7. Statistical Analysis

Descriptive parameters were expressed as the mean ± standard deviation (SD). Frequencies were presented as percentages. Chi square and Fisher’s exact tests were used as appropriate for categorical or ordinal variables. For continuous variables, *t* tests were used to compare two variables, and one-way analysis of variance (ANOVA) was used for more than two variables.

OS was calculated in months from the date of surgery to death (related or unrelated to cancer) or to the date of the last follow-up visit for surviving patients. CSS was calculated as the time from the date of surgery to cancer-related death, and DFS was calculated as the time from the surgery to cancer recurrence. Women who were alive and without recurrence were censored at the date of the last follow-up visit. The Kaplan–Meier method was used to estimate the survival distribution. Tick marks indicate censored data. The comparison test chosen for analysis of survival was the log-rank test. Effects were expressed as hazard ratios (HRs) with 95% confidence intervals (CIs). Cox proportional hazard models included established prognostic factors: age, FIGO status, lymph node metastasis, residual disease, Aletti’s SCS, and complications. In the multivariate analysis, missing data were considered as absent. A *p*-value lower than 0.05 was considered statistically significant.

## 3. Results

### 3.1. Characteristics of the Study Population

Of the 1171 women with ovarian cancer who received surgical treatment during the study period, 144 were excluded from analysis because of no surgical treatment (41 with only chemotherapy and seven patients without any curative treatment), a non-epithelial ovarian carcinoma or no FIGO stage reported. The distribution of the remaining 979 women in the three age groups was as follows: 615 women (62.8%) aged 65 years or younger, 225 women (22.6%) aged 65–74 years, and 139 women (14.2%) aged 75 years or older (Figure 1).

The mean age at diagnosis was 60.1 years (±12.7). The demographic and clinicopathologic characteristics of the whole cohort by age group are reported in Table 1. The number of comorbidities was higher in the older groups (70% of the elderly and 66% of the very elderly patients had three or more comorbidities vs. 23% of the young patients, *p* < 0.001). The ASA score was higher in the oldest group: 32/139 (23%) vs. 51/615 (8%) in the young group (*p* < 0.001). The tumor characteristics are reported in Table 1. The pathologic subtype was serous, endometrioid, mucinous, mixed and clear-cell for 66%, 11%, 5%, 12 and 6%, respectively, for the whole population. More women in the 65–74-year age group had serous ovarian cancer (vs. other epithelial tumors) (*p* = 0.048) (Table 1). Patients older than 65 years had more advanced FIGO stage (i.e., >IIa): 76% for <65-year group, 84% for 65–74-year group, and 80% for ≥75-year group (*p* = 0.033). There was no significant difference concerning the presence of lymph node metastasis (Table 2).

### 3.2. Surgical Procedures and Adjuvant Therapies

The surgical procedures are described in Table 2. Thirty-three percent of the elderly women (67/225) had primary cytoreductive surgery vs. 44% (242/625) of the young patients and 40% (45/139) of the very elderly (*p* < 0.05). According to the surgical procedure groups described by Luyckx [20], the very elderly had fewer standard procedures (40% (34/84) vs. 59% (104/177) for the elderly and 72% (384/530) for the young patients, (*p* < 0.001)). The very elderly patients had significantly fewer pelvic and para-aortic lymphadenectomies (49% for ≥75 years, 70% for 65–74 years, and 80% for <65 years, *p* < 0.001) with fewer lymph nodes removed among those who underwent a para-aortic lymphadenectomy (*p* < 0.05). The very elderly also had less upper-abdominal surgery (11%, 15/139) than the elderly (21%, 47/171) and young patients (23%, 139/615) (*p* < 0.05). Only 9% (13/139) of the very elderly had an Aletti’s SCS higher than eight compared with 16% and 22% for the elderly and young groups, respectively (*p* < 0.001). There was more residual disease in the two older groups (30%) than for younger women (20%) (*p* < 0.05).

Adjuvant treatments are displayed in Appendix A. No adjuvant chemotherapy was performed in 21% (24/139) of the very elderly patients, 14% (28/225) of the elderly and 13% (71/615) of the young patients (*p* = 0.084). When chemotherapy was conducted, the very elderly were administered fewer cycles (neoadjuvant plus adjuvant) compared with the elderly and the young: 23% of the very elderly received fewer than six cycles vs. 10% in the two other groups (*p* = 0.003). Only 12% (11/139) of the very elderly women received bevacizumab compared with 26% (112/615) of the young patients and 20% (30/225) of the elderly (*p* < 0.05).

### 3.3. Perioperative Complications

There was no difference in minor and major Clavien-Dindo complications in the three groups (*p* = 0.164 for minor complications and *p* = 0.567 for major complications). No significant difference in the rate of postoperative complications was observed (21.3% in the group <65 years, 20.9% in the group 65–74 years and 17.3% in the group ≥75 years) whether in terms of transfusion, digestive, urinary and pulmonary complications, complications of the abdominal wall, infections or repeat surgery. No patients died in the young group whereas one patient died postoperatively in each of the older groups (Table 3).

### 3.4. Survival Results

The mean follow-up was 36 months (±29.3). Recurrences were observed in 511 of the 979 women (52.2%) with a mean time of recurrence of 26.68 months (±24.7). In the whole population, the 3-year DFS was 44% (95% CI: 39.8–48.8), the 3-year CSS was 76.4% (95% CI: 72.4–80.5) and the 3-year OS was 71% (95% CI: 67.4–75.2) (Figure 2). Three-year DFS, CSS and OS rates for the three age groups are described in Table 3.

Univariate analysis for the 3-year OS showed that an age over 65 years, a FIGO stage III (HR = 3.702, 95% CI: 2.30–5.95) or IV (HR = 6.318, 95% CI: 3.70–10.77) (*p* < 0.001), residual disease (HR = 3.226, 95% CI: 2.51–4.15; *p* < 0.001) and lymph node metastasis (HR = 2.81, 95%CI: 1.91–4.12; *p* < 0.001) were significantly associated with a lower OS. The multivariate analysis is shown in Table 4.

## 4. Discussion

The present study highlights that women over 65 years old with ovarian cancer undergo less radical surgery due to higher comorbidities even though they present with more advanced FIGO stage disease. Surgical treatment was found to be less complex for the elderly, resulting in similar postoperative complications, whether minor or major, as in the younger patients. Similarly, for systemic treatment, the elderly were less likely to have six chemotherapy cycles or be administered a targeted therapy such as bevacizumab. These findings are concordant with those in the literature [8,11,13,16,22,23]. Thus, due to a higher advanced FIGO stage and less optimal treatment in elderly ovarian cancer patients, age is significantly correlated with poorer prognosis in patients with epithelial ovarian cancer, as prior studies have described [8,9,14,16,24].

The fact that elderly patients with more comorbidities have similar perioperative complications suggests that surgeons consider these patients to be frailer and perform less radical surgery to limit morbidity [25]. However, residual disease was an important prognostic factor in OS (HR = 3.226, 95% CI: 2.51–4.15; *p* < 0.001) independently of age [25], and the elderly could benefit from complete cytoreductive surgery. Decisions to adjust treatment protocols are often guided by chronological age, which is illustrated in clinical trials which tend to recruit young patients [26,27]. With 70% of complete cytoreduction surgery in the elderly and very elderly patients, our rate was higher than that described in literature (29.5% between 60 and 79 years old [5], and 21.7 to 25 % for patients older than 80 years old in the SEER database [11,13]). Based on the SEER database, Warren et al. reported that in an ovarian cancer patient over 75 years of age, an appropriate surgery, an appropriate chemotherapy, and both surgery and chemotherapy were only realized, respectively, 37.6% (OR = 0.58 (IC 95% 0.40–0.83)), 51.2% (OR = 0.27 (IC 95% 0.17–0.41)) and 18.9% (OR = 0.36 (IC 95% 0.28–0.530)) [28]. Cloven et al. showed that only 21.7% of patients older than 80 years of age were optimally cytoreduced (73.7% for patients < 60 years old and 29.5% between 60 and 79 years old) [13]. For Gershenson et al., patients older than 65 were correctly debulked half as often as younger patients (33% vs. 61%) [29].

This may explain the similar OS and CSS rates for both age groups older than 65 years and would seem to indicate that 65 years marks a threshold of two groups of ovarian cancer patients that require different guidelines of therapeutic management. However, adapting standardized treatment guidelines for women over 65 years with ovarian cancer is hampered by a lack of evidence-based medicine as elderly patients are historically under-represented in clinical trials [15] [30,31,32]. Frailty is a common finding in patients with ovarian cancer and is independently associated with worse surgical outcomes and poorer OS. Kumar et al. showed that frailty was independently associated with death (adjusted hazard ratio: 1.52, 95% CI: 1.21–1.92) after adjusting for known risk factors [33]. Median survival time was in favor of younger patients (98 vs. 30 months) and less frail patients (56 vs. 27 months) [34]. Although routine assessments of frailty can be incorporated into patient counseling and decision-making for the ovarian cancer patient beyond simple reliance on single factors such as age, there are no prospective studies on the effects of age or frailty on modifications to surgical approaches, postoperative complications, or prognosis in elderly women with ovarian cancer.

Adjuvant treatment in this population is also a subject of debate. In our study, very elderly patients received less adjuvant treatment than their younger counterparts and fewer cycles. Although several studies [13,35,36,37] have demonstrated that patients older than 70 years tolerate standard chemotherapy with similar rates of initial response and with an improved median OS of six months [38,39,40], the probability of receiving chemotherapy decreases with age [23] and the number of comorbidities [41]. There are currently minimal data evaluating the effectiveness and safety of bevacizumab in elderly women, but Amadio et al. showed that the occurrence of serious (grade ≥3) adverse events did not increase among the older group [42]. Besides, Perren and al. [43] demonstrated its effectiveness in women with residual disease or FIGO 4 status. As elderly patients are more likely to have residual disease, we believe that they could be good candidates for this therapy.

One of the strengths of our study is the high number of patients included from eight centers. However, our study has limitations that need to be considered when interpreting the data. We did not have validated cut-offs to categorize patients as “elderly” or “very elderly,” so comparison with other studies is difficult. However, there is no current consensus to define elderly women. We chose to classify women into three groups—“young” (<65 years), “elderly” (65–74 years), and “very elderly” (≥75 years)—whereas Yancik et al. [44] made the distinction between the “young” old (65–74 years), “older” old (75–84 years) and “oldest” old (≥85 years). Some reports chose a cut-off at 65 years old [38], others at 70 [37,45] or 75 years old [46], and some up to 80 years old [12,13]. Another limitation is linked to the retrospective nature of the study with incomplete data sets (such as ICU transfer, hospital stay, or unplanned 30 days). Furthermore, we did not know the reasons for cessation of chemotherapy, especially in the older age group. Of note, these points were evaluated in a previous publication which showed that hematological and cardiovascular toxicities were more frequent in elderly patients, but this did not influence prior discontinuation of therapy [47]. In addition, all the patients were managed in an expert center by surgeons specializing in oncologic gynecology who usually perform complete standard care (a combination of surgery and chemotherapy) [48]. Therefore, they might not be representative of all elderly women with ovarian cancer. In the same way, a common concern with observational data is the potential for selection bias, in which unobserved dimensions of health status, such as performance status, may determine treatment and independently affect survival as we described above. Indeed, the number of comorbidities was significantly higher in the elderly patients. Similarly, elderly patients who did not undergo lymphadenectomy received adjuvant treatment less often than the elderly patients who did (*p* = 0.07), implying that patient care is influenced, at least in part, by subjective evaluation of health status. The high burden of medical comorbidities, financial and geographic barriers to care, as well as patient preferences, may influence treatment and survival [36]. Nevertheless, similar to other studies, no objective evaluation was used to tailor surgical staging or adjuvant treatment. Lastly, patients were included before three milestone publications that led to modifications in national and international recommendations for the management of ovarian cancer in 2019 [49,50]: a reduced indication for lymphadenectomy in ovarian cancer because of the results of the LION (lymphadenectomy in ovarian neoplasms) study [51], the place of olaparib in patients with BRCA (breast cancer) mutations [52], and the use of hyperthermique intra-peritoneal chemotherapy (HIPEC) [53].

## 5. Conclusions

The incidence of ovarian cancer is increasing with the aging of the population. Currently, elderly and very elderly women are at risk of incomplete surgery with less upper abdominal surgery and more residual disease, despite having more aggressive ovarian cancer. These elements are prognostic factors for women’s survival regardless of age. Moreover, they also have less adjuvant therapy (chemotherapy and bevacizumab) because they are perceived as being too frail to tolerate these treatments. Elderly oncology patients should undergo individual oncogeriatric assessment to determine the risk/benefit balance of therapy. To do so, specific trials dedicated to the elderly should be performed to generate evidence-based medicine and guidelines for the management of elderly patients with cancer, specifically in ovarian cancer, which requires aggressive surgery and systemic treatment.

## Figures and Tables

**Figure 1 jcm-09-01451-f001:**
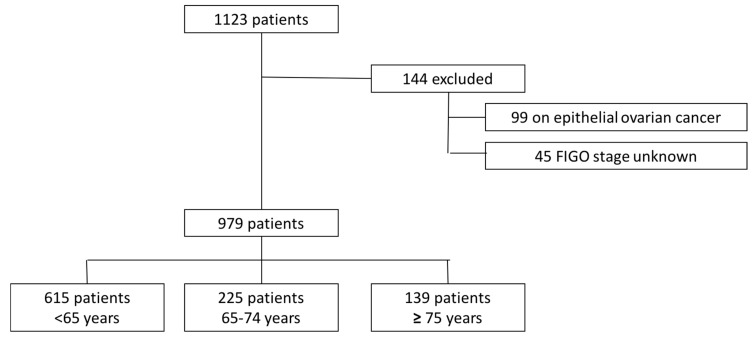
Flow-chart of patient inclusion.

**Figure 2 jcm-09-01451-f002:**
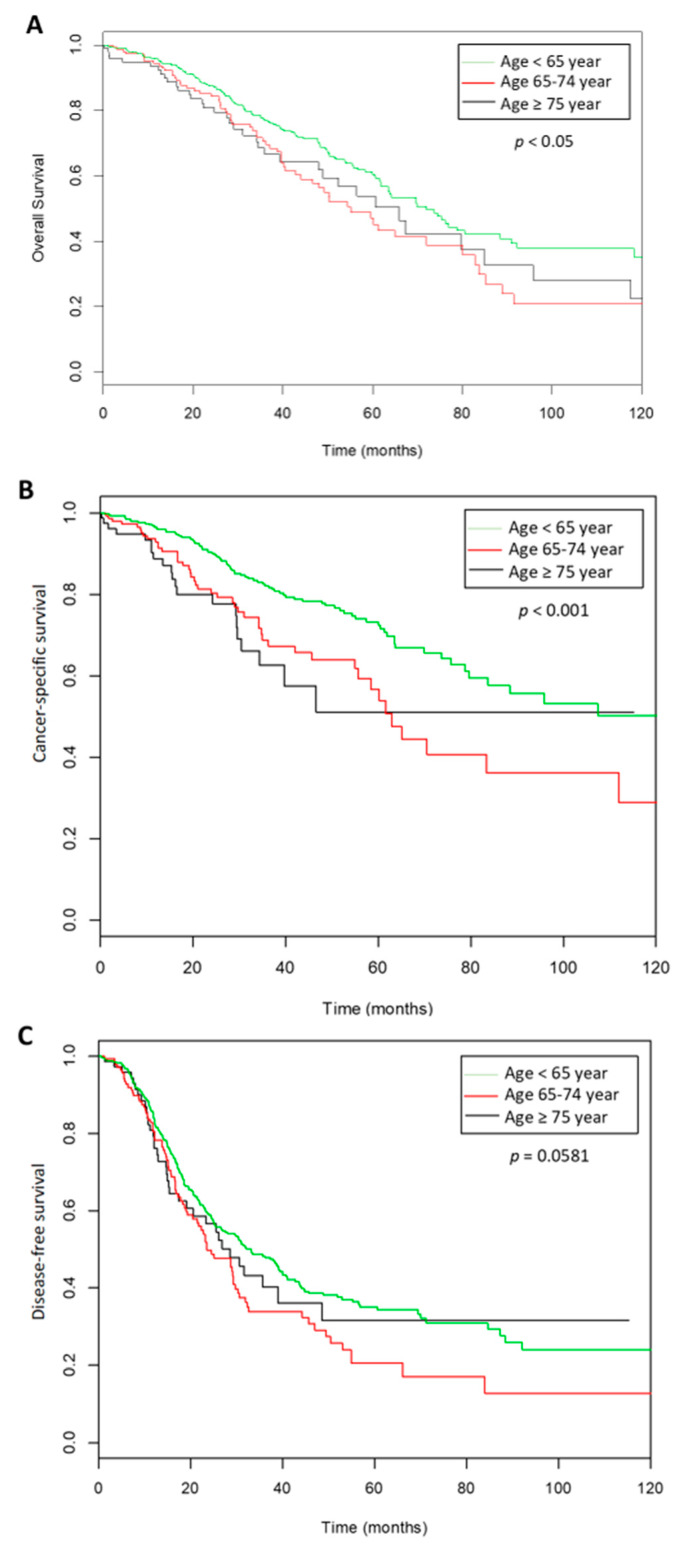
Survival curves according the three groups of ages. (**A**). Overall survival, (**B**). cancer specific survival, (**C**). disease free survival.

**Table 1 jcm-09-01451-t001:** Patient and Tumor Characteristics.

	Population		Age < 65		Age 65–74		Age ≥ 75		*p* Value
	*n* =	979	*n* =	615	*n* =	225	*n* =	139	
	*n* =	100%	*n* =	62.8%	*n* =	22.6%	*n* =	14.2%	
Parity (mean ± SD)	1.78	±1.49	1.58	±1.35	2.09	±1.71	2.16	±1.58	**<0.001**
Parity									
≤2	685	74%	460	79%	146	69%	79	61%	**<0.001**
>2	242	26%	124	21%	67	31%	51	39%	
NC	52		31		12		9		
Menopause									
Yes	718	78%	354	63%	225	100%	139	100%	**<0.001**
No	207	22%	207	37%	0		0		
NC	54		54		0		0		
Body Mass Index (mean ± SD)	25.12	±5.48	25.24	±5.76	25.3	±5.03	24.2	±4.84	0.875
Comorbidity									
≥3	277	39%	107	23%	109	70%	61	66%	**<0.001**
<3	428	61%	350	77%	46	42%	32	34%	
Diabetes									
Yes	41	4%	18	3%	16	7%	7	6%	**<0.001**
No	938	96%	597	97%	209	93%	132	94%	
Arterial Hypertension									
Yes	181	18%	74	12%	56	25%	51	37%	**<0.001**
No	798	82%	541	88%	169	75%	88	63%	
ASA score									
1–2	644	66%	402	65%	151	67%	91	65%	**<0.001**
3–4	112	11%	51	8%	29	13%	32	23%	
NC	223		162		45		16		
Ca125 at diagnosis (mean ± SD)	1620	±3983	1664	±4129	1512	±4031	1601	±3165	0.351
Pathologic type									
Serous	647	66%	394	64%	165	73%	88	63%	**0.048**
Vs other epithelial tumors	332	34%	221	36%	60	27%	51	37%	
FIGO stage (2018)									
≤IIa	211	22%	147	24%	36	16%	28	20%	**0.033**
>IIa	768	78%	468	76%	189	84%	111	80%	
Lymph node metastasis									
Yes	257	33%	194	37%	52	30%	11	12%	0.064
No	360	46%	249	48%	79	45%	32	34%	
NC	169		75		43		51		

NC: Non communicated; FIGO: International Federation of Gynecology and Obstetrics. Bold for significant value.

**Table 2 jcm-09-01451-t002:** Treatment Characteristics.

	Population		Age < 65		Age 65–74		Age ≥ 75		*p* Value
	*n* =	979	*n* =	615	*n* =	225	*n* =	139	
		100%		62.8%		22.6%		14.2%	
Diagnostic laparoscopy									
Yes	756	98%	483	98%	173	96%	100	97%	0.208
No	18	2%	8	2%	7	4%	3	3%	
NC	205		124		45		36		
Neoadjuvant Chemotherapy									
Yes	518	59%	309	56%	137	67%	72	62%	**0.028**
No	355	41%	243	44%	67	33%	45	38%	
NC	106		63		21		22		
Neoadjuvant Chemotherapy, number of preoperative cycles									
≤4	173	45%	115	50%	40	36%	18	42%	**0.039**
>4	209	55%	113	50%	71	64%	25	58%	
NC	136		81		26		29		
Para-aortic Lymphadenectomy									
Yes	614	74%	447	81%	128	69%	39	44%	**<0.001**
No	214	26%	108	19%	57	31%	49	56%	
NC	151		60		40		51		
PAo nodes removed (mean ± SD)	17.37	9.89	18.03	10.17	16.37	9.22	12.9	7.01	**0.033**
Pelvic Lymphadenectomy									
Yes	618	75%	455	82%	122	70%	41	48%	**<0.001**
No	201	25%	103	18%	53	30%	45	52%	
NC	160		57		50		53		
Pelvic nodes removed (mean ± SD)	15.50	10.00	15.71	10.27	14.99	9.36	14.9	9.22	0.101
Aletti’s score									
Low (≤3)	296	30%	143	23%	74	33%	79	57%	**<0.001**
Intermediate (4–7)	502	51%	339	55%	116	52%	47	34%	
High (≥8)	181	18%	133	22%	35	16%	13	9%	
Residual Disease									
R+	221	24%	118	20%	64	30%	39	30%	**0.038**
R0	715	76%	473	80%	152	70%	90	70%	
NC	43		24		9		10		
Total chemotherapy cycles									
<6	80	11%	46	10%	17	10%	17	23%	**0.003**
vs. ≥6	637	89%	434	90%	146	90%	57	77%	
NC	262		135		62		65		
Bevacizumab									
Yes	153	23%	112	26%	30	20%	11	12%	**0.033**
No	521	77%	321	74%	119	80%	81	88%	
NC	305		182		76		47		

PAo: para-aortic. R+: residual disease. R0: no residual disease. NC: Non communicated. Bold for significant value.

**Table 3 jcm-09-01451-t003:** Three-year disease-free survival, cancer-specific survival and overall survival rates (univariate analysis).

Characteristics	Disease Free Survival Rate, % (95%CI)	*p*	Cancer Specific Survival Rate, % (95%CI)	*p*	Overall Survival Rate, % (95%CI)	*p*
**Age <65 years**	47.2 (42.4–52.7)	*p* = 0.057	81.7 (77.3–86.3)	*p* < 0.001	76.2 (71.8–80.8)	*p* < 0.001
**Age 65–74 years**	33.7 (25.8–43.9)		67.3 (58.4–77.6)		61.3 (53.1–70.8)	
**Age ≥75 years**	37.2 (26.7–51.8)		57.4 (43–76.7)		56.4 (44.5–71.4)	

**Table 4 jcm-09-01451-t004:** Three-year disease-free survival, cancer-specific survival and overall survival rates (multivariate analysis).

	Disease Free Survival HR	*p*	Cancer-Specific Survival HR	*p*	Overall Survival HR	*p*
	(95% CI)		(95% CI)		(95% CI)	
**Age**						
<65	1		1		1	
65 to 74	0.998 (0.72–1.38)	0.988	1.29 (0.76–2.18)	0.343	1.29 (0.86–1.93)	0.214
≥75	0.77 (0.49–1.23)	0.277	1.742 (0.79–3.82)	0.166	1.33 (0.77–2.30)	0.31
**FIGO**						
1	1		1		1	
2	0.986 (0.42–2.33)	0.974	0.564 (0.06–5.17)	0.612	0.457 (0.10–2.11)	0.316
3	2.486 (1.43–4.32)	**0.001**	2.625 (0.87–7.94)	0.088	1.64 (0.80–3.38)	0.175
4	3.39 (1.76–6.51)	**<0.001**	4.90 (1.41–16.97)	0.012	2.23 (0.97–5.12)	0.059
**Time of surgery**						
Surgery after 6 cycles	1		1		1	
Primary debulking surgery	1.24 (0.30–5.12)	0.758	0.327 (0.07–1.47)	0.145	0.27 (0.08–0.84)	**0.025**
Surgery after 3 or 4 cycles	2.25 (0.55–9.16)	0.254	0.659 (0.16–2.77)	0.569	0.40 (0.13–1.25)	0.117
**Residual Disease**						
R0	1		1		1	
R1	0.974 (0.68–1.40)	0.88	1.26 (0.68–2.32)	0.459	1.74 (1.15–2.65)	**0.009**
**Aletti’s Score**						
<8	1		1		1	
≥8	1.456 (0.99–2.14)	0.057	1.129 (0.62–2.06)	0.692	1.47 (0.88–2.45)	0.137
**Adjuvant treatment**						
No	1		1		1	
Yes	0.308 (0.07–1.28)	0.10	0.59 (0.14–2.45)	0.468	1.23 (0.39–3.79)	0.723
**Lymph node metastasis**						
No	1		1		1	
Yes	1.76 (1.23–2.50)	**0.002**	1.99 (1.09–3.61)	**0.024**	2.27 (1.36–3.79)	**0.002**
**Complications according to Clavien-Dindo classification**						
No	1		1		1	
1	0.582 (0.38–0.88)	0.011	0.77 (0.41–1.44)	0.410	0.55 (0.32–0.94)	**0.029**
2	0.861 (0.54–1.37)	0.526	2.13 (1.15–3.93)	**0.015**	1.41 (0.82–2.42)	0.207
3	0.69 (0.36–1.34)	0.277	1.4 (0.49–4.02)	0.534	1.59 (0.71–3.54)	0.257
4	0.307 (0.12–0.77)	**0.011**	NA	NA	0.50 (0.15–1.67)	0.263

Bold means significant value.

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
