# Peer review of "Management and Survival of Elderly and Very Elderly Patients with Ovarian Cancer: An Age-Stratified Study of 1123 Women from the FRANCOGYN Group"

_jcm, 2020, doi:10.3390/jcm9051451_

Round 1

Reviewer 1 Report

Methods:

1) Were there any patients treated with chemotherapy only and excluded ?

2) Any data on patient functional status ?

3) Any data available on ICU transfer, hospital stay, unplanned 3--day readmission 

4) lines 110-111: I would recommend discussing here the rationale of selecting these cut-offs

5) lines 147-149: was follow-up uniform in all centers ? Was CA-125 routinely checked?

6) Line 151: please clarify what you mean by "surgical management" as primary outcome? Aggressiveness of debulking?

7) Was there any co-linearity in the Cox model between Aletti's score and complications?

8) Would recommend performing stratified analysis by stage, residual disease status and (PDS vs IDS) and evaluating survival. 

9) My major concern in the methodology is the inclusion of patients with stage II disease. I would recommend limiting your analysis to stage III-IV patients only since these patients require real debulking surgery 

Results

1) Table 3 is too busy, would recommend providing it as a supplemental table 

2) How was mean follow-up calculated ? Reverse Kaplan Meier method?

3) Table 5: seems that the model is fitted only among patients with available data for all variables. Please state how many patients were included in the final model 

4) I am confused regarding variable "time to surgery" since it includes interval debulking and surgery after 6 cycles, don't these two overlap since they represent NACT ? 

5) What was the distribution of Aletti's score in each age group?

Discussion:

1) I would recommend expanding your discussion and include surgical outcomes on elderly reported by other groups. 

2) Line 278-279: what was the rate of bevacizumab among R1 patients in each age group in your study ??

3) Agree that individual oncogeriatric assessment should be performed, I would comment in the discussion on the already available data 

Author Response

Authors thank reviewer for their clever comments that allowed to improve manuscript. See below answer for queries.

1°) The draft was edited by a native English, named Felicity Neilson, who worked in an editing company.

2°) Were there any patients treated with chemotherapy only and excluded?

It is right. Indeed, patients with only chemotherapy were excluded, as mentioned in material and methods (page 3, line 99). We replaced sentence “we excluded patients with no surgical treatment…” by “we excluded patients with no surgical treatment (thus only chemotherapy patients were excluded), …” . Furthermore, page 5, line 174, we added the number of patients without surgical treatment: “Of the 1171 women with ovarian cancer who received surgical treatment during the study period, 144 were excluded from analysis because of no surgical treatment (41 wil only chemotherapy, and 7 patients without any curative treatment) non-epithelial ovarian carcinoma or no FIGO stage reported.”

3°) Any data on patient functional status ?

Authors agree that no functional status data available is a pitfall of present study, but ASA score and number of comorbidities were reported. In discussion section, we reported this weakness (line 296, page 12) with the followed sentence “In the same way, a common concern with observational data is the potential for selection bias, in which unobserved dimensions of health status, such as performance status, may determine treatment and independently affect survival, as we described above.”

4°) Any data available on ICU transfer, hospital stay, unplanned 3--day readmission 

Authors agree that no data about ICU transfer, hospital stay, unplanned 3--day readmission were a pitfall of present study, but postoperative complications were well described in table 3. In discussion section, we reported this weakness (line 291, page 12) with the followed sentence “Another limitation is linked to the retrospective nature of the study with incomplete data sets (such as ICU transfer, hospital stay, unplanned 30 day readmission).”

5°) lines 110-111: I would recommend discussing here the rationale of selecting these cut-offs

Authors totally agree with this point, which is strongly discusses in discussion section (page 12, line 287): We did not have validated cut-offs to categorize patients as “elderly” or “very elderly” so comparison with other studies is difficult. However, there is no current consensus to define elderly women. We chose to classify women into three groups –“young” (<65 years), “elderly” (65–74 years), and “very elderly” (≥75 years)– whereas Yancik et al. [39]made the distinction between the “young” old (65–74 years), “older” old (75–84 years) and “oldest” old (≥85 years). Some reports chose a cut-off at 65 years old [34], others at 70 [33, 40]or 75 years old [41]up to 80 years old [12, 13]. »

6°) lines 147-149: was follow-up uniform in all centers? Was CA-125 routinely checked?

The patient follow-up was uniform in all centers because it was defined by nationwide recommendations with clinical evaluation each 4 months the first two years and every 6 months after. During inclusion period, the national recommendation did not recommend the use of Ca 125 screening for post-therapeutic follow-up.

7°) Line 151: please clarify what you mean by "surgical management" as primary outcome? Aggressiveness of debulking?

Authors agree with this comment. The outcome measures were: “The outcome measures were OS, DFS, and CSS rates calculated from date of recurrence, date of death, and date of cancer-related death”. The followed sentence was removed: “The primary outcome measure was the surgical management of the patients.”

8°) Was there any co-linearity in the Cox model between Aletti's score and complications?

It is a good question, but we did not find co-linearity between these two variables, probably due to modification of aggressiveness of debulking according age or frailty, subjectively assessed by surgeons.

9°) Would recommend performing stratified analysis by stage, residual disease status and (PDS vs IDS) and evaluating survival. 

Authors agree with this recommendation, but with new analysis, we have to add a lot of data tables that could confuse take home message. These new analyses were a new work that could lead to a new publication.

10°) My major concern in the methodology is the inclusion of patients with stage II disease. I would recommend limiting your analysis to stage III-IV patients only since these patients require real debulking surgery 

Authors agree with this comment, but authors would also show that ovarian cancer patient with surgical management have different ovarian cancer disease. Indeed, in table 1, we showed significant differences for FIGO stages or pathologic subtypes according age groups.

11°) Table 3 is too busy, would recommend providing it as a supplemental table 

As requested, the table 3 was provided as supplemental table.

12°) How was mean follow-up calculated? Reverse Kaplan Meier method?

The follow-up mean was calculated with each follow-up time (follow-up time defined as time between diagnosis and time of last contact of patient or death) of each patient of cohort.

13°) Table 5: seems that the model is fitted only among patients with available data for all variables. Please state how many patients were included in the final model 

The reviewer is right. We added a sentence in material and methods that indicated: “In the multivariate analysis, missing data were considered as absent”. Indeed, for the multivariate analysis, if a data was missed in the model we have to remove patient, that is why we have to remove some patient.

14°) I am confused regarding variable "time to surgery" since it includes interval debulking and surgery after 6 cycles, don't these two overlap since they represent NACT ? 

Reviewer is right. In table 4 (which is the old table 5), the term of “interval debulking surgery” was replaced by “Surgery after 3 or 4 cycles”.

15°) What was the distribution of Aletti's score in each age group?

The Aletti’s score in each age group was added in table 2.

16°) I would recommend expanding your discussion and include surgical outcomes on elderly reported by other groups. 

Authors agree. Surgical outcomes was expanding in discussion section using literature data. Thus, the following sentences were added in discussion section: “Based on SEER database, Warren et al reported that in ovarian cancer patient over 75 years of age, an appropriate surgery, an appropriate chemotherapy and both surgery and chemotherapy were only realized respectively 37.6% (OR=0.58 (IC 95% 0.40-0.83)), 51.2% (OR = 0.27 (IC 95% 0.17-0.41)) and 18.9% (OR = 0.36 (IC 95% 0.28-0.530)) [28]. Cloven et al. showed that only 21.7% of patients older than 80 years of age were optimally cytoreduced (73.7% for patients < 60 years old and 29.5% between 60 and 79 years old) [13]. For Gershenson et al, patients older than 65 were correctly debulked half as often as younger patients (33 vs 61%) [29]. “

17°) Line 278-279: what was the rate of bevacizumab among R1 patients in each age group in your study ??

Bevacizumab was always used in patients with R+, thus the rate of bevacizumab among R1 patients was similar in each age group as the rate of R+ patients in each age group (table 2).

18°) Agree that individual oncogeriatric assessment should be performed, I would comment in the discussion on the already available data 

Authors totally agree with this point of view. A section was added about this point in discussion: Frailty is a common finding in patients with ovarian cancer and is independently associated with worse surgical outcomes and poorer OS. Kumar et al showed that frailty was independently associated with death (adjusted hazard ratio: 1.52, 95% CI: 1.21-1.92) after adjusting for known risk factors [33]. Median survival time was in favor of younger patients (98 versus 30 months) and less-frailty patients (56 vs 27 months) [34]. Although routine assessments of frailty can be incorporated into patient counseling and decision-making for the ovarian cancer patient beyond simple reliance on single factors such as age, there are no prospective studies on the effects of age or frailty on modifications to surgical approaches, postoperative complications, or prognosis in elderly women with ovarian cancer.

Reviewer 2 Report

This manuscript enhances clinical knowledge about elderly patients with ovarian cancer and is well written. I have only a few comments to add/discuss:

  • Do you have more data of the comorbidities? What were the most frequent comorbidities? Was there are difference between the different age groups?
  • Do you also have data on comedications and amount of comedications/polypharmacy?
  • Do you have data on ECOG performance status that you could include in your analyses?
  • Regarding the Kaplan Meier curves why do you think the group of 65-74 y olds have worse survival compared to the pts > 75 years?
  • The rate of patients having received standard treatment according to guidelines is not very high - was there a specific selection procedure or geriatric assessment before surgery? Can you discuss the low rate of standard procedures?
  • Do you know the reasons for prior cessation of chemotherapy? What were the reasons for early cessation in the older age group? Were there differences in reasons between the age groups? There is a publication by Woopen et al. at Eur J Cancer 2016 showing that there are more toxicities due to chemotherapy in elderly patients and discussing early cessation of chemo in elderly compared to younger patients. Do you also have data on toxicities? Were toxicities a major reason for early cessation?
  • Regarding data on bevacizumab in elderly (page 12 lines 277/278) - within the Otilia study there were analyses for elderly patients that were presented at the German Cancer Conference in 2020 by Sehouli et al. (abstract 684)

I recommend to accept this manuscript for publication after addressing the mentionned comments.

Author Response

Authors thank reviewer for their clever comments that allowed to improve manuscript. See below answer for queries.

1°) The draft was edited by a native English, named Felicity Neilson, who worked in an editing company.

2°) Do you have more data of the comorbidities? What were the most frequent comorbidities? Was there are difference between the different age groups?

Authors had data on diabetes and Arterial hypertension in table 1 (characteristics of patient). The most frequent morbidities were AHT and diabetes. There are a lot of kind of comorbidities, but it is heterogeneous and difficult do summarize in single table.  

3°) Do you also have data on comedications and amount of comedications/polypharmacy?

Sorry, but authors did not have any data on medications.

4°) Do you have data on ECOG performance status that you could include in your analyses?

Authors agree that no functional status data available is a pitfall of present study, but ASA score and number of comorbidities were reported. In discussion section, we reported this weakness (line 296, page 12) with the followed sentence “In the same way, a common concern with observational data is the potential for selection bias, in which unobserved dimensions of health status, such as performance status, may determine treatment and independently affect survival, as we described above.”

5°) Regarding the Kaplan Meier curves why do you think the group of 65-74 y olds have worse survival compared to the pts > 75 years?

Authors agree with reviewer. We showed that the group of 65-74 y olds have worse survival compared to the pts > 75 years with survival analysis reported in table 3. The Kaplan meier survival curves showed that the threshold seems 65 years of age. We argued this in discussion with the followed sentences: This may explain the similar OS and CSS rates for both age groups older than 65 years and would seem to indicate that 65 years marks a threshold of two groups of ovarian cancer patients that require different guidelines of therapeutic management.

6°) The rate of patients having received standard treatment according to guidelines is not very high - was there a specific selection procedure or geriatric assessment before surgery? Can you discuss the low rate of standard procedures?

The rate of RO surgery is high when we compared with literature: higher than 70% for the whole cohort. For chemotherapy, more than 81% of patient had at least 6 cycles of chemotherapy, that was the standard of treatment of advanced ovarian cancer (table 2). Thus, more than 75% of the cohort had standard procedures according French recommendation. There is no specific or geriatric assessment before surgery.

7°) Do you know the reasons for prior cessation of chemotherapy? What were the reasons for early cessation in the older age group? Were there differences in reasons between the age groups? There is a publication by Woopen et al. at Eur J Cancer 2016 showing that there are more toxicities due to chemotherapy in elderly patients and discussing early cessation of chemo in elderly compared to younger patients. Do you also have data on toxicities? Were toxicities a major reason for early cessation?

Authors agree with reviewer, but we did not have data about reason why medical team stopped chemotherapy. It was always because of toxicity and low tolerance of chemotherapy regimens, but the exact reason (haematological toxicity or other toxicities …) was not reported. We added this pitfall in discussion section. The followed sentence was added in discussion: “Furthermore, we did not know the reasons for cessation of chemotherapy, especially in the older age group. Of note, these points were evaluated in previous publication, which showed that hematological and cardiovascular toxicities were more frequent in elderly patients, but this did not influence prior discontinuation of therapy [46] » (page 12, line 336).

8°) Regarding data on bevacizumab in elderly (page 12 lines 277/278) - within the Otilia study there were analyses for elderly patients that were presented at the German Cancer Conference in 2020 by Sehouli et al. (abstract 684)

Authors totally agree with reviewer. The followed sentence was added in discussion section: “There are currently few data evaluating the effectiveness and safety of bevacizumab in elderly women, but Amadio et al showed that the occurrence of serious (grade ≥3) adverse events did not increase among the older group [42].” This reference was the analysis of OTILIA German non-interventional study on behalf of the North-Eastern German Society of Gynaecological Oncology

Round 2

Reviewer 1 Report

-revised manuscript incorporated suggested changes